

# Molecular and serological dynamics of *Chlamydia pecorum* infection in a longitudinal study of prime lamb production

Sankhya Bommana[1], Evelyn Walker[2], Marion Desclozeaux[1], Martina Jelocnik[1], Peter Timms[1], Adam Polkinghorne[1] and Scott Carver[3]

[1] Centre for Animal Health Innovation, University of the Sunshine Coast, Sippy Downs, Australia
[2] Central West Local Land Services, Dubbo, Australia
[3] School of Biological Sciences, University of Tasmania, Hobart, Tasmania, Australia

## ABSTRACT

**Background**. *Chlamydia pecorum* is a globally significant livestock pathogen causing pathology and production losses. The on-farm infection and serological dynamics and the relevance of existing diagnostic tools for diagnosing *C. pecorum* in livestock remains poorly characterized. In this study, we characterized the antigen and antibody dynamics of this pathogen in a longitudinal study of prime lamb production, utilizing the infection focused *C. pecorum*-specific 16S rRNA qPCR assay and serology based chlamydial Complement fixation Test (CFT).

**Methods**. The study consisted of 76 Border Leicester mixed sex lambs (39 females and 37 males) that were sampled bimonthly from 2–10 months of age in a commercial farm operating in Central NSW, Australia. Blood/plasma was analysed for CFT antibodies, and swabs from conjunctival, rectal and vaginal sites were analysed for *C. pecorum* shedding using qPCR. We assessed the temporal and overall dynamics of *C. pecorum* in lambs, including detailed description and comparison of qPCR and CFT, the timing of first detection by either diagnostic method, the lag between infection and antibody response; and the distribution of qPCR load and CFT antibody titre over time.

**Results**. Over the study period, *C. pecorum* was highly prevalent (71.0% by qPCR, 92.1% by CFT, 96.0% by both), with 21.1% (16/76) lambs shedding $\geq 1,000$ qPCR copies/$\mu$l (denoted as high shedders). *C. pecorum* shedding (as evidence of infection) were first observed at two months of age (14.4%) with a significant peak of infection occurring at six months of age (34.2%), whereas seroconversions peaked at eight months of age (81.5%). 52.6% of *C. pecorum* qPCR and CFT positive lambs became qPCR negative by 10 months of age, indicating clearance of chlamydial infection. Although CFT is utilised for on-farm detection of active infection, we confirm that it lagged behind qPCR detection (average lag $1.7 \pm 2.1$ months) and that the proportion of qPCR positives simultaneously identified by CFT was low with 2/11 (18.1%), 0/13, 17/25 (68.0%), 5/7 (71.4%) and 1/10 (10.0%) concurrent seroconversions occurring at two, four, six, eight and 10 months of age, respectively.

**Discussion**. This work reveals rapid rates of *C. pecorum* infection and widespread exposure during lamb production. The comparison of molecular and serological diagnostic agreement longitudinally, supports the use of qPCR as an important ancillary tool for the detection of active infections in conjunction with chlamydial CFT for

Corresponding author
Scott Carver, scott.carver@utas.edu.au

routine veterinary diagnostics. Development of rapid Point-of-Care (POC) tools for diagnosing active infection would be valuable for producers and veterinarians.

# INTRODUCTION

Despite growing evidence that infections of the obligate intracellular bacterial pathogen, *Chlamydia pecorum,* are ubiquitous among some of the most economically important livestock species globally, the epidemiology and pathologies associated with this pathogen are poorly understood (*Walker et al., 2015*). *C. pecorum* infections of sheep, cattle and goats are associated with polyarthritis (*Walker et al., 2016*), keratoconjunctivitis (*Polkinghorne et al., 2009*), sporadic bovine encephalomyelitis (SBE) and pneumonia (*Jelocnik et al., 2014a*). Sporadic cases of ovine and caprine abortions due to *C. pecorum* have also been reported (*Giannitti et al., 2016*; *Walker et al., 2015*). *C. pecorum* chlamydiosis in both sheep and cattle can limit growth and survival of young rapidly growing stock and, such weight loss or failure to thrive as a result of *C. pecorum* polyarthritis, is the primary economic concern for farmers (*Poudel et al., 2012*; *Walker et al., 2015*; *Walker et al., 2016*). The latter disease is a compelling one for Australian producers with 2.1% of lambs and 1.6% of calves condemned each year at Australian abattoirs as a result of polyarthritis, estimated to cost the livestock industry around $30M annually (*Walker et al., 2016*). Similar economic costs of arthritis have also been reported elsewhere (*Dupuy et al., 2013*). The specific contribution of arthritis-associated losses by *C. pecorum* are yet to be established, however.

The complex relationship between *C. pecorum* infection and overt animal pathology makes the diagnosis and control of infections challenging. Sub-clinical, asymptomatic infections are common, characterised by the detection of *C. pecorum* in the faeces, gastrointestinal and/or urogenital tract of so-called "shedder" animals (*Reinhold et al., 2008*; *Reinhold, Sachse & Kaltenboeck, 2011*). These same animals may also act as an important reservoir, facilitating infection of individuals who exhibit symptomatic infections. Indeed, in Australia, the largest exporter of sheep globally, *C. pecorum* was recently estimated to be present in 30% of the country's sheep flock, based on faecal shedding alone (*Yang et al., 2016*). While these infections are common, it is apparent that *C. pecorum* can also disseminate to other tissues where it can replicate in epithelial cells and macrophages of the conjunctival, genital and intestinal sites, in synoviocytes of the joint tissue and occasionally, the respiratory tract (*Jelocnik et al., 2014a*; *Twomey et al., 2006*). The factors that influence dissemination and pathogenesis of these strains are currently unknown, although molecular typing studies have suggested that genetic differences may exist between strains associated with disease and those found asymptomatically colonising the gastrointestinal tract (*Jelocnik et al., 2013*; *Jelocnik et al., 2014b*; *Mohamad et al., 2014*).

In Australia, diagnosis of sheep chlamydiosis is based on clinical history, symptoms and presenting pathology, and is routinely confirmed by diagnostic laboratories using

a Complement Fixation Test (CFT). Chlamydial CFT detects antibodies (Ab) to either whole chlamydial elementary bodies (EBs) or *Chlamydiaceae*-specific lipopolysaccharide and remains as the recommended test for *Chlamydia* diagnosis by the World Organisation for Animal Health and Sub-Committee for Animal Health Laboratory Standards (*Sachse et al., 2009*). The use of crude or partially purified antigen in CFT depends on the binding of anti-*Chlamydiaceae* antibodies of the host species to guinea pig complement, and has highly variable sensitivity depending on the host species and antibody isotype (*Kaltenboeck et al., 1997*; *Perez-Martinez, Schmeer & Storz, 1986*). Moreover, use of whole chlamydial elementary bodies (EBs) or *Chlamydiaceae*-specific lipopolysaccharide as an antigen in this assay renders only genus-specificity and inevitable serological cross-reactivity with *Chlamydia*-related organisms and gram negative bacteria (*Casson, Entenza & Greub, 2007*; *Haralambieva et al., 2001*). Detection of sero-conversion by CFT has multiple purposes such as: (a) confirmation of chlamydiosis; (b) the presence or absence of chlamydial infection; and (c) determination of immune status after vaccination (*Sachse et al., 2009*). The major issue is that CFT has a largely unknown relationship to either the acute, convalescent or persistent phase of *C. pecorum* infection or even pathology itself (*Griffiths et al., 1996*; *Kaltenboeck et al., 1997*; *Perez-Martinez, Schmeer & Storz, 1986*). Alongside this unknown relationship, *C. pecorum* shedding (measured by qPCR) at the gastrointestinal tract has not well correlation to disease or pathology and has not been investigated further to date (*Walker et al., 2016*).

Beyond the use of CFT and qPCR for the diagnosis of *C. pecorum* cell culture or embryonated hens' egg based isolation methods exist that are often cumbersome and time consuming for on-farm settings (*Sachse et al., 2009*). Here we conducted a longitudinal study of *C. pecorum* infection in a prime lamb flock. Our aims were to: (1) describe the temporal and overall dynamics of *C. pecorum* in lambs, including detailed description and comparison of qPCR and CFT, the timing of first detection by either diagnostic method, and estimate the lag between infection (qPCR) and antibody response (CFT); and (2) detail the distribution of chlamydial qPCR load and CFT antibody titre over time. For both aims, we detail consistency and variation in infection across three anatomical sampling sites: conjunctiva, rectum and vagina.

## MATERIALS AND METHODS

**Lamb husbandry.** This study followed 76 Border Leicester mixed sex lambs operating in a commercial sheep farm in Central Western NSW. Lambs in this study were managed as per normal farm husbandry practices (e.g., marking, feeding, weaning, vaccinations etc.). At time of marking (two months of age), the lambs were uniquely ear tagged and sampled at bi-monthly intervals (two, four, six, eight and 10 months) until finishing which is at the 10 months of age.

**Blood and swab samples collection.** Swab samples taken from conjunctiva, rectum and vagina, and blood samples were collected from individual lambs across five bimonthly sampling time points from 2–10 months of age. Collected swabs were used for in-house DNA amplification assays, and the blood samples were submitted to the State Veterinary
Diagnostic Lab, Elizabeth Macarthur Agricultural Institute, Menangle, NSW, for serological testing by *Chlamydia* CFT. A serum sample was considered positive by CFT with a titer of 16 or greater in this study. The collection and testing of these swabs and blood samples was approved by the University of the Sunshine Coast Animal Ethics Committee (AN/S/14/31).

**Swab processing and DNA extraction.** Clinical swabs were processed according to the in-house swab processing methods (*Jelocnik et al., 2013*). Briefly, swabs were dispensed into 1.5 ml of sucrose-phosphate-glutamate buffer by vortexing and centrifugation. The resulting cell pellet was resuspended in tris-EDTA buffer and heated at 95 °C for 10 min to heat-inactivate the elementary bodies (EBs) for further processing at room temperature. For all swabs, DNA was extracted using a QIAamp DNA kit (Qiagen, Doncaster, Victoria, Australia), according to the manufacturer's instructions. DNA purity and yield was determined using a NanoDrop spectrophotometer ND-1000 (Thermo Fisher Scientific, Inc., Waltham, MA, USA).

*C. pecorum* **16S species- specific qPCR screening.** Conjunctival, vaginal and rectal swabs were screened for the presence of *C. pecorum* infections by a *C. pecorum*-specific quantitative PCR targeting the 16S rRNA gene modified from previously published protocol (*Marsh et al., 2011*). The *C. pecorum* 16S 204 bp fragment (RT-Cpec -F: 5′-AGTCGAACGGAATAATGGCT-3′, RT-Cpec-R: 5′-CCAACAAGCTGATATCCCAC-3′; IDT) was sub-cloned into pGem-T Easy (Promega, Madison, WI, USA) and amplified with M13 universal primers to generate a M13-Cpec-16S fragment. Serial dilutions of the M13-Cpec-16S fragment were used to produce a standard curve by mixing 5 µl of diluted fragment with RT-Cpec-F and RT-Cpec-R primers (1 µM final) and 1X QuantiTect SYBR® Green PCR mix (Qiagen, Doncaster, Victoria, Australia) in a final volume reaction of 20 µl. Cycling conditions were 95 °C- 15 min, followed by 35 cycles of 94 °C- 15 s, 57 °C-15 s, 72 °C- 30 s, and a final amplification cycle of 72 °C, 10 min. Diluted *C. pecorum* strain MC/Marsbar served as a positive control while $dH_2O$ was used as negative control. All samples were tested in duplicates. The limit of detection load was 10 *C. pecorum* 16S rRNA copies/µl.

## Statistical analysis

We undertook a detailed individual assessment of the timing and diagnostic characteristics of *C. pecorum* infection dynamics. This included the diagnostic infection and serological response (qPCR and CFT) status at each sampling time point (Table 1), information on evidence of individual clearance of infection (as determined by a lamb having both a qPCR and CFT diagnosis across the sampling time points, but only being CFT positive by 10 months of age). This individual level information facilitated flock-wide description of *C. pecorum* infection dynamics and diagnostic evaluations, including overall and time dependent CFT and qPCR positivity and the diagnostic states (qPCR+CFT+, qPCR+CFT-, qPCR-CFT+, qPCR-CFT-) (Table 2). We evaluated the 'qPCR CFT agreement' as the percentage of qPCR positive and negative lambs with matching CFT diagnosis.

$$\frac{(a+d)}{(a+b+c+d)} \times 100$$
**Table 1** Characteristics of *C. pecorum* positivity of lambs (*n* = 76), as tested by qPCR at each sampling site (conjunctiva, rectum and vagina) and CFT.

| Age (months) | qPCR+ | | | | | CFT+ | |
|---|---|---|---|---|---|---|---|
| | Conj. No. | Rect. No. | Vag. No. | Combined No. | Combined % | No. | % |
| 2 | 3 | 5 | 3 | 11 | 14.4 | 10 | 13.1 |
| 4 | 12 | 4 | 2 | 18 | 21 | 1 | 1.3 |
| 6 | 10 | 9 | 11 | 21 | 34.2 | 42 | 55.2 |
| 8 | 5 | 3 | 5 | 13 | 15.7 | 62 | 81.5 |
| 10 | 9 | 1 | 2 | 12 | 15.7 | 25 | 32.8 |
| Overall | 39 | 22 | 23 | 54 | 71.0 | 70 | 92.1 |

**Table 2** Characteristics of qPCR CFT agreement, and the % of lambs that cleared their infection by 10 months of age (as determined by a lamb having both a qPCR and CFT diagnosis across the sampling time points, but only being CFT positive by 10 months of age).

| Age (months) | qPCR+ CFT+ No. | qPCR+ CFT- No. | qPCR- CFT+ No. | qPCR- CFT- No. | qPCR CFT agreement % | Infection clearance % |
|---|---|---|---|---|---|---|
| 2 | 2 | 6 | 8 | 60 | 81.5 | _ |
| 4 | 0 | 16 | 1 | 59 | 77.6 | _ |
| 6 | 22 | 4 | 23 | 27 | 64.4 | _ |
| 8 | 9 | 3 | 53 | 11 | 26.3 | _ |
| 10 | 5 | 7 | 20 | 44 | 64.4 | _ |
| Overall | 51 | 3 | 19 | 3 | 71.0 | 52.6 (*n* = 40) |

where *a* is diagnostic state qPCR+CFT+, *b* is qPCR+CFT-, *c* is qPCR-CFT+, and *d* is qPCR-CFT-. We have specifically avoided the widely used 'sensitivity' and 'specificity' terminology owing to the knowledge that qPCR and CFT are measuring inherently different things (bacterial agent and antibodies, respectively). The comparison of CFT against qPCR is nevertheless functional on applied grounds, owing to its relatively common use in veterinary settings to diagnose active infection.

Focussing only on the month at which first qPCR diagnosis was made, individual lags of CFT from qPCR diagnosis was made followed by average ± standard deviation to get the population/flock lag. Lag in CFT from qPCR was calculated on the basis of animals that had their qPCR detection first followed by/concurrent CFT detection (*n* = 45) (Tables S1–S3). Lambs that had CFT detection arise before qPCR (*n* = 31) as a result of previous infections that were missed due to bimonthly sampling frequency were not included in the lag analyses.

All data analyses was performed in Microsoft Excel 2013 and graphs outputs (Figs. 1 and 2) generated using GraphPad Prism version 7 (GraphPad Software, LaJolla, CA, USA).

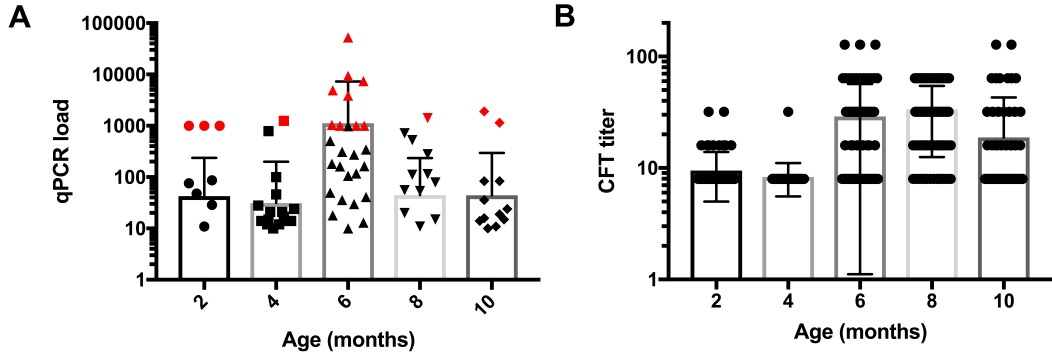

**Figure 1** **Age dependent distribution of *C. pecorum* bacterial load and antibody titers (A) qPCR load, lambs shedding high loads of *C. pecorum* DNA ($\geq 1,000$ copies/µl) and CFT positive are indicated as data points in red (B) CFT titer.** From 2–6 months of age, lambs exhibited an overall increase in mean chlamydial loads, peaking at six months of age ($1,128 \pm 6,205$). Mean CFT titres were low at 2–4 months of age ($9.5 \pm 4.5$ and $8.3 \pm 2.8$ at two and four months), and then peaked from 6–8 months of age ($29 \pm 27.8$ and $33.5 \pm 21$ at six and eight months), followed by a slight decline in mean titre at 10 months of age ($18.8 \pm 24$), respectively.

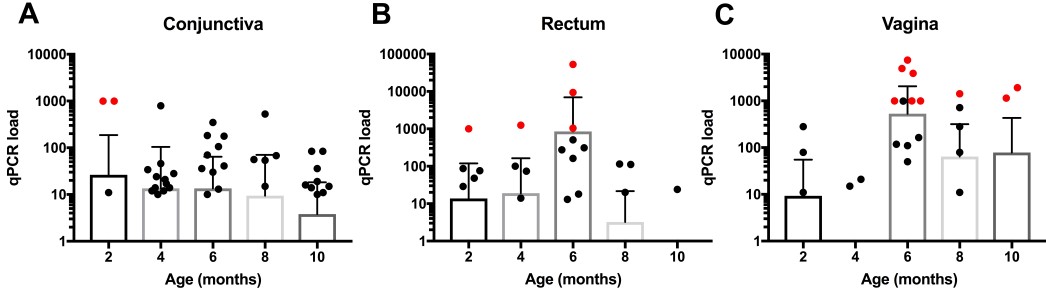

**Figure 2** **Age dependent distribution of *C. pecorum* bacterial load at (A) Conjunctiva (B) Rectum and (C) Vagina.** Lambs shedding high loads of *C. pecorum* DNA ($\geq 1,000$ copies/µl) and CFT positive are indicated as data points in red. The mean conjunctival bacterial loads trended downward from 2–10 months of age ($26.5 \pm 161.1$ to $3.8 \pm 14.4$, respectively), in contrast, mean bacterial loads increased from 2–6 months of age ($16.3 \pm 115.3$ to $849.6 \pm 6,143$) at the rectal sampling site, diminishing markedly thereafter ($0.8 \pm 2.3$ at 10 months). For female lambs the greatest bacterial loads occurred from 6–10 months of age ($544.2 \pm 1,538$ and $78.3 \pm 352.3$ at six and 10 months) at the vaginal sampling site.

## RESULTS

### Temporal and overall dynamics of *C. pecorum* exposure in lambs

Diagnosis of *C. pecorum* infections revealed an overall positivity of 71.0% by qPCR, 92.1% by CFT and 96.0% by either test over the study period (Table 1). The temporal patterns of qPCR detection were broadly consistent among the samples from the conjunctiva, rectum and vagina, with greatest number of detection at six months of age and similar among other months of sampling (Table 1). While temporal patterns in qPCR were similar among sampling sites, greater numbers of detection were made from the conjunctiva, relative to

**Table 3  First detection of *C. pecorum* in lambs at each sampling site (conjunctiva, rectum and vagina) by qPCR (infection).**

| Age (months) | qPCR+ (first detections) | | | | |
|---|---|---|---|---|---|
| | Conj. No. | Rect. No. | Vag. No. | Combined No. | Combined % |
| 2 | 3 | 5 | 3 | 11 | 14.4 |
| 4 | 9 | 2 | 2 | 13 | 17.1 |
| 6 | 10 | 5 | 10 | 25 | 32.8 |
| 8 | 3 | 2 | 2 | 7 | 9.2 |
| 10 | 7 | 1 | 2 | 10 | 13.1 |

the rectum and vagina (Table 1). Detection of *C. pecorum* by CFT rose rapidly from six months of age, peaking at eight months of age and then declining by 10 months of age (Table 1). A notable number of CFT detection ($n = 10$) were made at two months of age, and possibly indicate the presence of maternal antibodies, or infections that occurred and resolved <two months of age. Temporal agreement between qPCR and CFT was greatest at two months of age, decreased to lowest levels at eight months of age (owing to an increased number of lambs that were positive by CFT, but negative by qPCR), and increased again by 10 months of age (owing to an increased number of lambs negative by both diagnostic methods) (Table 2). Approximately half of the lambs (52.6%; 40/76) sampled exhibited evidence of infection clearance (only CFT detection by 10 months of age) within the sampling frame of this study (Table 2).

In this study, 67.1% of lambs (51/76) tested positive by both CFT and qPCR (Table 2). However, 3/51 of these lambs had CFT antibodies arise before qPCR detection due to past infection that went undiagnosed as a result of bimonthly sampling or detection of maternal antibodies at two months of age (Table S1). Interestingly, 3.9% (3/76) of lambs were qPCR positive and CFT negative throughout the study period and 2/3 of these lambs had recurrent *C. pecorum* infection as measured by repeated qPCR positivity (Table 2). We identified a further 25% (19/76) of lambs that tested positive by CFT and negative by qPCR during the entire study period of two to 10 months of age, and 3.9% (3/76) of lambs tested qPCR and CFT negative during the entire study period (Table 2).

We also investigated the timing of first detection by qPCR and CFT to gain an improved understanding of when lambs first became infected and seroconverted (Tables 3 and 4). Peaks in timing of first detection by qPCR and CFT were six and eight months of age, respectively (Tables 3 and 4), which was broadly similar to what was observed when evaluating the flock as a whole (Table 1). Most of the first seroconversions (CFT diagnoses) were detected concurrently with first detection of infection, a smaller number lagged behind infection and a moderate number occurred without detection of infection, which may indicate the 'window of infection' that was missed due to the bi-monthly sampling intervals (Tables 3 and 4). Overall, the sampling intervals in this study led to an estimated average time lag between first qPCR detection and first CFT detection of $1.7 \pm 2.1$ months.

**Table 4   First detection of *C. pecorum* in lambs by CFT (antibodies) in relation to age.** Lambs with their first CFT detection were further categorised into (a) new CFT detection due to concurrent seroconversion i.e., matching qPCR detection, (b) new CFT detection due to delay in seroconversion as a result of previous qPCR positive infection, and (c) new CFT detection due to suspected previous infection that was missed due to bi-monthly sampling.

| Age (months) | CFT+ No. | CFT+ % | CFT+ due to seroconversion | CFT+ due to lag in seroconversion | CFT+ due to suspected previous exposure |
|---|---|---|---|---|---|
| 2 | 10 | 13.1 | 2 | 0 | 8 |
| 4 | 0 | 0 | 0 | 0 | 0 |
| 6 | 30 | 39.4 | 17 | 5 | 8 |
| 8 | 20 | 26.3 | 5 | 9 | 6 |
| 10 | 1 | 1.3 | 1 | 0 | 0 |

**Table 5   Characteristics of *C. pecorum* bacterial load (qPCR load) and CFT antibody titers from 2 to 10 months of age.** The qPCR load and CFT titers are expressed as mean and standard deviation.

| Age (months) | qPCR combined Mean | qPCR combined SD | Conjunctiva Mean | Conjunctiva SD | Rectum Mean | Rectum SD | Vagina Mean | Vagina SD | CFT Mean | CFT SD |
|---|---|---|---|---|---|---|---|---|---|---|
| 2 | 42.8 | 195.9 | 26.5 | 161.1 | 16.3 | 115.3 | 9.5 | 46.4 | 9.5 | 4.5 |
| 4 | 31.6 | 169.4 | 13.5 | 91.1 | 19.03 | 144.7 | 0.9 | 4.1 | 8.3 | 2.8 |
| 6 | 1,128 | 6,205 | 13.3 | 50.7 | 849.6 | 6,143 | 544.2 | 1,538 | 29 | 27.8 |
| 8 | 44.8 | 192.1 | 9.6 | 61.3 | 3.4 | 18.4 | 64.3 | 253.8 | 33.5 | 21 |
| 10 | 44.3 | 253.6 | 3.8 | 14.4 | 0.3 | 2.8 | 78.3 | 352.3 | 18.8 | 24 |

## Distribution of *C. pecorum* bacterial load and antibody titre in lambs

*C. pecorum* bacterial load (qPCR copy number) and CFT antibody titer exhibited distinct temporal patterns (Table 5, Figs. 1A and 1B). From 2–6 months of age, lambs exhibited an overall increase in chlamydial loads, peaking at six months of age (Table 5, Fig. 1A). Across all time points, a small number of lambs had loads exceeding 1,000 16S rRNA gene copy numbers (Fig. 1A, data points in red). The temporal patterns in bacterial load varied among the sampling sites (Fig. 2). In the conjunctiva, loads trended downward from 2–10 months of age (Fig. 2A). In contrast, bacterial loads increased from 2–6 months of age at the rectal sampling site, diminishing markedly thereafter (Fig. 2B). For female lambs the greatest loads occurred from 6–10 months of age at the vaginal sampling site (Fig. 2C). Overall, 16/76 (21%) of lambs were shedding high loads of *C. pecorum* DNA ($\geq$1,000 copies/$\mu$l) from their mucosal sites of conjunctiva (3/76), rectum (4/76) and vagina (9/76), respectively (Fig. 2). CFT titres were low at 2–4 months of age, and then peaked from 6–8 months of age, followed by a slight decline in titre at 10 months of age (Table 5, Fig. 1B).

## DISCUSSION

Despite the significant economic loss in livestock due to chlamydiosis, *C. pecorum* infections continue to be under recognized as a major endemic pathogen for producers globally (*Walker et al., 2015*; *Walker et al., 2016*). Key factors contributing to this

underestimation are the lack of appreciation of infection dynamics, relative to antibody dynamics (*Berri et al., 2009*; *Sachse et al., 2009*). To address these issues, the objective of our study was to describe infection (measured by a species specific 16S qPCR assay) and serological (measured by CFT) dynamics using a longitudinal on-farm sheep flock setting in Central NSW, Australia.

Whether detected by qPCR, CFT or both diagnostic methods, the overwhelming majority of lambs were exposed to *C. pecorum* at some point over the study period, indicating rapid rates of infection, and widespread exposure in the flock. Based on the sampling frequency employed, we observed that lambs experienced peak chlamydial infections over a period of 2–6 months of age followed by seroconversion at 6–8 months of age, and occasional detection of low-level recurrent chlamydial infection at 10 months. The sudden rise in infection prevalence at six months potentially reflects a response to the stress of marking, and associated herding of lambs into yards, increasing contact between infected lambs, and precipitating spread of chlamydial infections (*Poudel et al., 2012*; *Stanley & Jones, 2003*). Marking or mulesing has been recorded as the peak time for risk of infections by Erysipelas, the other major cause of bacterial arthritis in lambs (*Robson, 2003*). Peak seroconversion of lambs occurred at eight months of age, although a subset of lambs had circulating antibodies as early as two months, likely owing to maternal transfer of antibodies or early seroconversion in cases that were also qPCR positive. At early time points (two and four months), qPCR preceded CFT in detecting new infections, with a peak in coincidental first detection by both methods occurring at six months. Based on these findings, qPCR performs well at detecting infections early in life (2–6 months), new infections, and potentially recurrent infections. Antibody responses appeared to confer a degree of protection, as indicated by animals becoming qPCR negative in 52.6% (40/76) of cases by 10 months. However, our results indicated that the protection was not long lasting, as both declining seroprevalence, waning titres from 8–10 months of age and absence of CFT titres in some cases was observed, and may have contributed to repeat infections detected during these months ($n = 8$ and $n = 6$ at eight and 10 months, respectively).

CFT is periodically utilized for diagnosis of active *C. pecorum* infection in veterinary settings, hence it was important to characterize its association to qPCR diagnosis. We found that using CFT potentially obscures the ability to detect acute active chlamydial infections with 9/11 (81.8%), 13/13 (100.0%), 8/25 (32.0%), 2/7 (28.5%) and 9/10 (90.0%) qPCR detection preceding seroconversion at two, four, six, eight and 10 months of age, respectively. This important finding of poor matching between CFT and qPCR is similar to results reported in other *Chlamydia* studies (*Bas et al., 2001*; *McCauley et al., 2007*; *Persson & Boman, 2000*; *Sachse et al., 2009*). Overall, our results show that CFT has limited value for diagnosis of acute active infections. In this study, we also identified three lambs that were qPCR positive (2/3 lambs were repeatedly qPCR positive) but had no detectable CFT antibodies longitudinally. In our previous study on characterisation of humoral immune responses to naturally occurring *C. pecorum* infections we have identified moderate antigen specific MOMP and PmpG IgG antibodies present in these three lambs longitudinally (*Bommana et al., 2017*). These findings elude us to believe that these animals are not passive shedders of *C. pecorum* and that in reality CFT as an assay has relatively

poor sensitivity. CFT has also been criticized for its requirement of technical expertise, subjective interpretation, and its unsuitability for testing large numbers of specimens; when testing *C. abortus* infections in sheep (*McCauley et al., 2007*; *McCauley et al., 2010*). Conversely, our species-specific qPCR assay appears a much improved tool for early and active stages of infection (*Sachse et al., 2009*). CFT as a diagnostic assay is suitable for seroprevalence surveys and serological dynamic studies (such as this study) as it can detect antibodies to clinically unapparent chlamydial infections. CFT is inappropriate for the retrospective diagnosis of chlamydial mucosal infections, such as in oculo-genital sites that make antibodies specific to a localised site of infection (*Barnes, 1989*; *Griffiths et al., 1996*). The exception to this is diagnosis of chlamydial polyarthritis or abortion in ruminants, wherein high exposure to *Chlamydia* elicits a pronounced systemic increase in antibody levels (*Perez-Martinez, Schmeer & Storz, 1986*; *Bommana et al., 2017*). Potentially, there may exist early seroconversion or antibody markers for improved serological diagnosis of acute-phase infection in livestock, overcoming the shortcomings of CFT. Future research is needed to identify such markers by investigating proteomic profiles of chlamydial infections to differentiate responses related to acute infections, cured past infection, and persistent infection. Improved diagnostic tools for acute phase infection, or exposure, that are simple to undertake and interpret, would be of significant value for routine use by veterinarians and veterinary diagnostic laboratories (*Jelocnik et al., 2017*).

A notable proportion (21.1%; 16/76) of the flock were shedding high numbers ($>1 \times 10^4$–$10^5$) of *C. pecorum*, which may have important implications for *C. pecorum* dynamics. "Supershedders" are suspected to facilitate transmission of many pathogens (*Lloyd-Smith et al., 2005*; *Matthews et al., 2013*). In this study lambs shedding high amounts of *C. pecorum* may facilitate transmission through fecal-oral contact or greater environmental contamination (e.g., water troughs, grazing pasture) (*Stanley & Jones, 2003*). Rapid POC diagnostic tools underway for *C. pecorum* and *C. psittaci*, (*Jelocnik et al., 2017*) are needed to identify these individuals as part of a preventative flock management program. Any animal that is identified early as a supershedder could be culled to avoid further flock transmission and reduce infection rates. More broadly, this observation fits with a body of literature suggesting that a small proportion of individuals are commonly responsible for the majority of transmission events (the 80:20 rule for supershedders and superspreaders) across multiple systems (*May & Anderson, 1987*; *Woolhouse et al., 1997*). One similar example was recorded in the livestock industry when faecal sample analysis conducted at a UK abattoir revealed that approximately 9% of the cattle examined over a 9-week period were high *Escherichia coli* O157 shedders ($>10^4$ CFU/g) and they accounted for over 96% of the bacteria shed by all animals tested (*Omisakin et al., 2003*).

## CONCLUSION

In this study, we have described the on-farm molecular and serological dynamics of the enigmatic pathogen, *C. pecorum* in lambs from two to 10 months of age. Lambs exhibiting *C. pecorum* infections had high rates of infection at six months of age and seroconversion at eight months of age. Based on our individual diagnostic assessments of qPCR and

CFT, *C. pecorum*-specific qPCR was found to be a more useful tool in diagnosing acute active infections early on in life (2–6 months), recurring infections, and also in detecting *C. pecorum* positivity in CFT negative animals. Further integration of the findings in this study with detailed pathological investigations in relation to chlamydial shedding, disease and impact are currently underway. Findings in this study provide insights into the infection and antibody dynamics, and ways to improve diagnosis of *C. pecorum* infections in sheep. While CFT is the primary assay in diagnosing chlamydial infections of livestock in veterinary settings, *C. pecorum*-specific qPCR assay will serve as an important ancillary tool when used in parallel with CFT in diagnosing active acute on farm infections of this pathogen. Further development of rapid POC assays for routine veterinary diagnostic testing of *C. pecorum* infections in Australian livestock would be valuable (*Jelocnik et al., 2017*). This is particularly the case for clinical cases with variable presentations, such as conjunctivitis, asymptomatic infections and/or conjunctivitis with polyarthritis. Considering the ever increasing availability of species and strain specific qPCR assays for member species in the genus *Chlamydia*, sensitive discriminatory assays for *C. pecorum* would be beneficial for both producers and researchers (*Jelocnik et al., 2017*; *Sachse et al., 2009*). Such tools will be essential if we are to truly understand the true prevalence and impact of these infections on livestock production globally.

## ACKNOWLEDGEMENTS

We would like to acknowledge Central West Local Land Services District Veterinarians and Biosecurity Officers for their on farm assistance in this project.

### Funding

This work was funded by an Australian Research Council Linkage Grant (LP140100315) awarded to Adam Polkinghorne, Peter Timms, Scott Carver and Evelyn Walker. The funders had no role in study design, data collection and analysis, decision to publish, or preparation of the manuscript.

### Grant Disclosures

The following grant information was disclosed by the authors:
Australian Research Council Linkage Grant: LP140100315.

### Competing Interests

The authors declare there are no competing interests.

### Author Contributions

- Sankhya Bommana performed the experiments, analyzed the data, wrote the paper, prepared figures and/or tables, reviewed drafts of the paper.
- Evelyn Walker conceived and designed the experiments, performed the experiments, reviewed drafts of the paper.

- Marion Desclozeaux and Martina Jelocnik performed the experiments, reviewed drafts of the paper.
- Peter Timms conceived and designed the experiments, contributed reagents/materials/ analysis tools, reviewed drafts of the paper.
- Adam Polkinghorne conceived and designed the experiments, contributed reagents/materials/analysis tools, wrote the paper, reviewed drafts of the paper.
- Scott Carver conceived and designed the experiments, analyzed the data, wrote the paper, prepared figures and/or tables, reviewed drafts of the paper.

### Animal Ethics

The following information was supplied relating to ethical approvals (i.e., approving body and any reference numbers):

The collection and testing of these swabs and blood samples was approved by the University of the Sunshine Coast Animal Ethics Committee (AN/S/14/31).

### Data Availability

The raw data is provided in Tables 1 and 2, and as a Supplemental File.

### Supplemental Information

Supplemental information for this article can be found online at http://dx.doi.org/10.7717/peerj.4296#supplemental-information.

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
