# Peer review of "Molecular and serological dynamics of Chlamydia pecorum infection in a longitudinal study of prime lamb production"

_PeerJ, doi:10.7717/peerj.4296_

## Round 0.1 · original submission · Minor Revisions

Dear Dr. Bommana and colleagues:

We have received reviews of your work and I anticipate accepting your work for publication after minor revision of the original manuscript. Importantly, please carefully process the many suggestions by Reviewer 3, especially how the results are presented in Table 1 and Table 2.

Likewise, Reviewer 1 feels that important explanation of data is missing in Figure 1 and Figure 2. Please also try to resolve the confusion associated with the qPCR terminology.

I look forward to your revision, good luck!

-joe

·

Basic reporting

Chlamydia pecorum is an important, and globally significant livestock pathogen causing pathology and production losses. The authors in this study applied molecular and serological tools to investigate the age-dependent distributions of C. pecorum in a lamb production. The manuscript is well written, and conclusions are well supported by the data. This is an important and well-designed study, and significantly improves our understanding on the C. pecorum dynamics.

Experimental design

Well-designed study.

Validity of the findings

The findings are novel.

Comments:

For table-3, the data are shown as mean,max and SD. Normally, we show the data as mean with SD, or mean with range (max, min). The authors may consider to modify the data.

Additional comments

1. Beyond qPCR and CFT used in this study, we have also culture and ELISA approaches for diagnosis of Chlamydia. The authors may address briefly the rationale to use only PCR and CFT in this study.

2. The authors should address clearly the specificity of CFT against C. pecorum alone; In the lamb production, it is not unusual to see the presence of other Chlamydia species;

3. Please reduce the length of the Statistical Analysis if possible;

4. The readers will benefit if the results for statistical analysis can be provide in the figures or legends of Figures 1 and 2;

Reviewer 2 ·

Basic reporting

Well written, but a little too many self citations in the introduction. Would be nice to see corrected.

Experimental design

no comment

Validity of the findings

no comment

Additional comments

Nice, clear, well written manuscript. No doubt about the validity and importance of the findings.

·

Basic reporting

The authors have presented a longitudinal study comparing detection of Chlamydia pecorum in lambs from 2 to 10 months of age via qPCR and CFT and discussed the relative merit of both techniques in terms of monitoring the presence of Chlamydia pecorum over time and as tools to guide the management of disease within a flock.

Longitudinal studies are hugely beneficial to the study of diseases especially those caused by intracellular pathogens. The study design has been well thought out and executed. The structure of the article is satisfactory and the literature cited is appropriate and sufficient and supports a strong knowledge of the topic.

The abstract, introduction and material and methods are well written. However, material and methods needs more detail (see below). The results, including tables 1 and 2, need a major revision. I think the authors have attempted to extrapolate more from their data than is available. It is clear that the authors have established interesting patterns within in and between their tests but I think they have overstated the interpretation of these results with respect to the progression of disease

Tables 1 and 2 are extremely confusing and should be reconsidered and reformatted. The authors have presented to much data in each table and should consider revising to make it easier for the reader. For example:

Table 1. A total number of animals qPCR+ at each time point, regardless of repeat positives from different samples, would aid in making sense of the comparative columns. Also, the 68 at the bottom of the second last column is a number not a percentage and equates to the 96% positive by both tests stated in the text. The totals at the bottom of columns 2-4 and the # footnote make no sense.

It might be worth having two tables, the first with the expanded explanation of the qPCR results and then these summarised to number of animals and compared to CFT in a second table.

Table 2. As with Table 1, this table is very confusing and should be revised as described in Table 1. The combination of numbers and percentages, time points and new detection permutations is not easy to follow. Once again, consider two tables as described for Table 1.

I am also concerned about the terminology used to describe the inferences from the qPCR results. Infectious load should be bacterial load as there are no references to how infectious load has been determined in the paper. A qPCR positive result is an indication of shedding but how does this correlate to the progression of disease without other information. The authors have also indicated the presence of “Supershedders” in the flock and animals that were qPCR positive but CFT negative. Could these animals have chlamydia simply passing through theire system following ingestion of faecal material from the “Supershedders” The “Supershedders” were eluded to a source of infection but not passive transmission.

I would encourage the authors to revise and simplify their results and the interpretation of results in accordance with what is achievable with qPCR and the progression of disease in the absence of other information, most notably pathology. At the very least some indication on the quantitative nature of the qPCR results needs to be correlated to disease if they choose not to revise.

Suggest revising the use of the term detections throughout

Suggest standardising infection load" "qPCR load" and "qPCR copy load" throughout

Line 52; Suggest Point-of-Care (POC) first and then POC throughout

Line 100: Has qPCR been correlated any better

Line 191 Delete, qPCR and CFT detection, with

Line 193: Revise sentence "meaning these lambs tested C. pecorum DNA qPCR positive and also seroconverted and developed a detectable antibody response measured by CFT"

Lines 197-200: "Interestingly, 3.9% (3/76) of lambs were qPCR positive and CFT negative throughout the study period and 2/3 of these lambs had recurrent C. pecorum infection as measured by repeated qPCR positivity" Surely this raises questions about the correlation between immune response, infection and shedding? Or the accuracy of the CFT test.

Line 273: unapparent

Line 474: Delete second use of word detections

Line 477: qPCR+ not PCR+

Experimental design

I think this is a new and valuable contribution to this field of research and would be valued by readers when revised.

The aim is well defined and the results, when revised, will make a valuable contribution to field of research and progress to contributing to the gap in knowledge.

In describing how qPCR was performed the authors refer to Marsh et al. 2011. There are several sets of primers in this reference and the set used here should be either shown in full or referred to by name with respect to the original reference. No mention is made of the qPCR reagents and should be included.

A description of how the bacterial load was determined quantitatively and what this means should also be included.

Generally speaking it would appear that the study has been undertaken to an acceptable standard and that all ethical requirements have been met.

Validity of the findings

Overall the study is quite straight forward. The data when revised will be a sound foundation to build on with future research in this area, particularly the application of qPCR and CFT, for monitoring the presence of chlamydia in research trials and as management tools on farm. this should be emphasized.

The discussion has been adequately written but will also improve with the revision of the results and their interpretation. As will the conclusion.

Whilst a revision of the results and toning down of speculation of these with regard to the progression of disease is warranted. The results do support differences in the dynamics of qPCR and serological dynamics of sheep exposed to Chlamydia pecorum.

The study should be published and will contribute to the knowledge of Chlamydia pecorum in prime lamb production

Additional comments

You have a great story, don't over complicate it. Be consistent in the use of terminology and be mindful of the use of terminology with respect to presence/absence of pathogens and how far this can be used to speculate on the progression of disease. Disease studies usually integrate pathology and histopathology to confirm findings.

---

## Round 0.2 · accepted · Accept

Dear Dr. Carver and colleagues:
I am delighted to inform you that your work is now suitable for publication in PeerJ. Congratulations! I believe this will be a valuable contribution to the field, and look forward to seeing it in print.
Best,
-joe